# Prognostic Impact of Dihydropyrimidine Dehydrogenase Germline Variants in Unresectable Non-Small Cell Lung Cancer Patients Treated with Platin-Based Chemotherapy

**DOI:** 10.3390/ijms24129843

**Published:** 2023-06-07

**Authors:** Javier Guijarro-Eguinoa, Sara Arjona-Hernandez, Stefan Stewart, Olga Pernia, Pedro Arias, Itsaso Losantos-García, Tania Rubio, Miranda Burdiel, Carlos Rodriguez-Antolin, Patricia Cruz-Castellanos, Oliver Higuera, Alberto M. Borobia, Sonia Rodriguez-Novoa, Javier de Castro-Carpeño, Inmaculada Ibanez de Caceres, Rocio Rosas-Alonso

**Affiliations:** 1Clinical Pharmacology Department, La Paz University Hospital, 28046 Madrid, Spain; 2Laboratory Medicine Department, Puerta Del Mar University Hospital, 11009 Cadiz, Spain; 3Cancer Epigenetics Laboratory, Genetics Department, La Paz University Hospital, 28046 Madrid, Spain; 4Experimental Therapies and Novel Biomarkers in Cancer, Hospital La Paz Institute for Health Research—IdiPAZ, 28029 Madrid, Spain; 5Pharmacogenetics Laboratory, Genetics Department, La Paz University Hospital, 28046 Madrid, Spain; 6Biostatistics Department, Hospital La Paz Institute for Health Research—IdiPAZ, 28029 Madrid, Spain; 7Oncology Department, La Paz University Hospital, 28046 Madrid, Spain; 8Genetics of Metabolic Diseases Laboratory, Genetics Department, La Paz University Hospital, 28046 Madrid, Spain

**Keywords:** pharmacogenetics, platinum resistance, cisplatin, carboplatin, non-small cell lung cancer, *DPYD*

## Abstract

Platin-based chemotherapy is the standard treatment for patients with non-small cell lung cancer (NSCLC). However, resistance to this therapy is a major obstacle in successful treatment. In this study, we aimed to investigate the impact of several pharmacogenetic variants in patients with unresectable NSCLC treated with platin-based chemotherapy. Our results showed that *DPYD* variant carriers had significantly shorter progression-free survival and overall survival compared to *DPYD* wild-type patients, whereas DPD deficiency was not associated with a higher incidence of high-grade toxicity. For the first time, our study provides evidence that *DPYD* gene variants are associated with resistance to platin-based chemotherapy in NSCLC patients. Although further studies are needed to confirm these findings and explore the underlying mechanisms of this association, our results suggest that genetic testing of *DPYD* variants may be useful for identifying patients at a higher risk of platin-based chemotherapy resistance and might be helpful in guiding future personalized treatment strategies in NSCLC patients.

## 1. Introduction

Platin (Pt) compounds are widely used and approved for the treatment of malignant tumors worldwide. Cisplatin, the first generation of Pt-based anti-cancer drugs, has been used as a chemotherapy agent since the 1970s, when it was first approved by the FDA for the treatment of testicular cancer [1,2], and it has since been key in the treatment of many types of cancer. Cisplatin exerts its antineoplastic effects by creating abnormal bonds between the Pt metal center and the DNA strands. These bonds, known as adducts, interfere with the normal functioning of DNA, leading to cell cycle arrest and ultimately apoptosis in proliferating cancer cells [3]. Cisplatin, therefore, has a therapeutic effect on several malignant solid tumors such as testicular, ovarian, head and neck, bladder, lung, cervical cancer, melanoma and lymphomas [1].

However, cisplatin is a chemotherapeutic drug that lacks specificity, resulting in systemic toxicity and the harming of healthy cells along with tumor cells, having a large range of unwanted side effects and dose-limiting toxicity, particularly nephrotoxicity, neurotoxicity, ototoxicity and myelosuppression [4]. When combined, these side effects can significantly diminish the quality of life of treated patients, requiring a reduction in the drug dosage or even an early discontinuation of treatment, which can ultimately compromise its efficacy. To address these issues, carboplatin, a second-generation Pt chemotherapy drug, was developed based on cisplatin and approved in 1989. Although the adducts formed in DNA by cisplatin and carboplatin are identical, the rate of adduct formation during exposure to carboplatin is much slower, and higher concentrations are required to produce the same number of adducts as cisplatin. Carboplatin has, therefore, demonstrated reduced systemic toxicity compared to cisplatin and can be used as a high-dose chemotherapy agent for aggressive tumors [5]. Despite this, cisplatin still remains the agent of choice for many aggressive tumors [6]. Nevertheless, drug resistance remains a significant concern during Pt chemotherapy and can limit the long-term effectiveness of both cisplatin and carboplatin [7]. Thus, researchers continue to study Pt-based compounds and develop new complexes with improved efficacy and safety profiles [8].

One of the main indications for Pt-based compound use is lung cancer, which remains one of the most frequently diagnosed cancers and the primary cause of cancer-related deaths [9]. In 2020 alone, primary lung cancers caused nearly 1.8 million deaths and accounted for over 2.2 million newly diagnosed cancer cases worldwide [10]. Non-small cell lung cancer (NSCLC) is responsible for more than 80% of lung malignancies, and, unfortunately, due to the insidious onset of symptoms, many patients are diagnosed at late stages [11]. Until recently, Pt-based chemotherapy and its combination with radiotherapy was the only standard systemic treatment considered for advanced stages of NSCLC. The advent of targeted therapies and immunotherapies over the last decade has dramatically changed the therapeutic approach for NSCLC [12]. Currently, the use of cisplatin and carboplatin in combination with novel therapies has emerged as a novel treatment standard [13,14]. For example, the addition of pembrolizumab to Pt-based therapy has shown significantly longer overall survival (OS) and progression-free survival (PFS) than chemotherapy alone, and, as a result, this approach is the new standard treatment in NSCLC [15,16,17].

The important variability in both the toxicity and effectiveness of Pt-based treatments among patients is a complex issue, and many pharmacogenomic studies have been conducted with the goal of identifying genetic alterations that can accurately predict how a patient will respond to these treatments. The findings from these studies are ushering in a new era of precision medicine for the treatment of NSCLC. An important example is the identification of genetic variants in *TPMT* that increase the risk of cisplatin-induced hearing loss in the pediatric population. In this regard, the Canadian Pharmacogenomics Network for Drug Safety clinical recommendation group has published guidelines for the use of pharmacogenetic testing for *TPMT* gene variants when prescribing cisplatin in pediatric cancer patients [18]. Other genes have been associated with outcomes such as nephrotoxicity and cardiotoxicity [19,20,21]. Several genetic variations have additionally been associated with increased and decreased likelihoods of a response to treatment using Pt-compounds [22,23].

Although sufficient evidence has been established to create guidelines for cisplatin use in pediatric patients based on the *TPMT* genotype, the limited findings available at the moment concerning other genes and outcomes are hindering the development of other guidelines to further personalize Pt-based therapy. Therefore, the aim of this study is to explore how certain genetic variants may affect the outcome of patients with NSCLC treated through Pt-based therapy.

## 2. Results

### 2.1. Patients’ Characteristics

In total, 55 patients diagnosed with NSCLC between October 2015 and August 2016 and treated with Pt-based chemotherapy were recruited. Patient data were collected retrospectively during 2019. The male–female ratio was 2:1, with an average age of 67 years (range of 47–82). All the patients had received Pt-based treatment during their disease. The characteristics of the patients at baseline were generally similar in the two groups (Table 1).

In our study population, 81.8% of the patients who received cisplatin-based therapy and 81.8% of those who received carboplatin-based therapy had at least 1 adverse event of any grade and of any kind. In total, 31.8% of the patients who received cisplatin-based therapy and 30.3% of those who received carboplatin-based therapy had an adverse event of grade >2. The most common adverse events in the 2 groups were gastrointestinal effects (68.2% of the patients who received cisplatin-based therapy and 48.5% of those who received carboplatin-based therapy), hematological effects (36.4% and 27.3%, respectively), the most frequent of which was anemia (22.7% and 15.2%, respectively) and other effects (59.1% and 66.7%, respectively). No statistically significant differences were found between patients treated with cisplatin-based therapy and patients treated with carboplatin-based therapy in terms of toxicity or efficacy, according to the 3 month follow-up Computed Tomography (CT) scan.

In terms of molecular analysis, the genes with the most variants were *UGT1A1* (52.83%), *MTHFR* (52.83%), *CYP2C19* (51.85%), *LY96* (47.17%), *EPAS1* (44.44%), *CYP2C9* (40.74%), *TFAP2B* (30.91%), *CYP2C8* (29.09%), *CYP3A5* (18.52%), *CYP3A4* (13.2%), *DPYD* (11.1%), *TPMT* (9.1%), *GM2A* (5.5%) and *NUDT15* (1.8%).

### 2.2. Efficacy Outcomes

As mentioned before, there were no significant differences regarding efficacy between the treatment groups. Of the cisplatin-treated patients, 13.6% of patients had progression detected in the 3-month follow-up CT scan from the start of treatment whereas 18.2% of carboplatin-treated patients had progression detected. Given the absence of discernible differences in efficacy between the groups, subgroup analysis was not performed.

PFS during Pt-based therapy was evaluated, according to the genetics polymorphism identified. This approach identified significant differences in PFS, according to the mutational status of the *DPYD* gene. This was not the case for the other genes, where no statistically significant differences were found between the wild-type genotype group and carrier group in terms of PFS.

The median Pt-based therapy PFS was 2.1 months (95% CI, 1.6 to 2.7) in the *DPYD* variants carrier group, versus 7.5 months (95% CI, 6.0 to 9.3) in the *DPYD* wild-type group. The hazard ratios for disease progression were 0.057 (95% CI, 0.015 to 0.210; *p* < 0.001) (Figure 1a). Therefore, having the wild-type *DPYD* genotype reduces the risk of progression by 94.3% compared to the carrier group.

The OS during Pt-based therapy was additionally evaluated, according to the polymorphism found. The median OS during Pt-based therapy among carriers of any variant in *DPYD* was 4.0 months (95% CI, 2.3 to 5.6) versus 15.4 months in the wild-type *DPYD* group (95% CI, 11.8 to 19.0) (hazard ratio for death, 0.18; 95% CI, 0.06 to 0.5; *p* = 0.001) (Figure 1b).

In our cohort, 6 patients (11.1%) carried variants in the *DPYD* gene (4 patients carried the c.1236G>A variant, and the other 2 carried the c.2846A>T variant). All of them experienced disease progression in the first reassessment CT scan, after the initiation of treatment. Furthermore, all of them died within 6 months of diagnosis (1 at 2 months, 2 at 4 months and 2 at 6 months), except for 1 of them who responded very favorably to second line immunotherapy treatment (pembrolizumab monotherapy).

In addition, the Pt-based therapy OS according to *CYP2C8* mutation status was near statistical significance. The median OS was 12.3 months (95% CI, 6.0 to 9.3) in the carrier *CYP2C8* group; however, statistical significant differences were not reached in the *CYP2C8* wild-type group (*p* = 0.066) (Figure 2). We found no differences between the OS and mutational status in other genes.

### 2.3. Toxicity Outcomes

The majority of patients with a variant in any of the genes studied developed toxicity, with alterations in the *GM2A*, *CYP3A4*, *LY96*, *UGT1A1* and *TFAP2B* genes present in 66.7%, 42.9%, 36%, 35.7% and 35.3% of patients with grade ≥ 2 toxicity reactions, respectively. However, no statistical significance was found between the presence of a variant in the genes studied and the development of toxicity.

No statistically significant differences were observed regarding toxicity within cisplatin and carboplatin treatment groups, as 81.8% of patients in each group developed toxicity. A minimal increase in the percentage of patients developing grade 2 or more toxicity was observed in cisplatin-treated patients, in comparison with carboplatin-treated patients (31.8% vs. 30.3%, respectively). Given the absence of differences in toxicity between the groups, subgroup analysis was not performed.

Among patients carrying a variant in the *DPYD* gene, the most frequent adverse reactions were gastrointestinal manifestations, such as constipation (grade 1), diarrhea (grade 1–2), nausea (grade 1) and vomiting (grade 1). A total of 3 of them experienced asthenia (grade 2–3), and only 2 experienced haematological toxicity: anaemia (grade 1) and thrombopenia (grade 2), respectively. However, none of these manifestations led to a discontinuation of treatment. Finally, 1 of them suffered liver failure in relation to progression, detected in the 3-month re-evaluation CT, requiring hospitalization. The patient died a few days later.

Results of the associations between pharmacogenetic variants and treatment-related toxicity are shown in Table 2.

## 3. Discussion

Although Pt-based chemotherapy remains a standard therapy for various types of cancer, resistance to these agents is a major clinical challenge. The treatment can involve an adaptive response within malignant cells that reduces their susceptibility to the antiproliferative and cytotoxic effects of the therapy, resulting in a resistance to treatment and ultimately to disease progression [24]. Several mechanisms of resistance have been identified, including a decreased drug uptake, increased drug efflux following an overexpression of ATP-binding cassette transporters, enhanced DNA repair mechanisms, alterations in apoptotic pathways or increased levels of reactive oxygen species, which can lead to oxidative stress and promote cancer cell survival [25,26].

It has been proposed that the variability in treatment responses observed among individuals with the same clinical characteristics may be attributed, at least in part, to genetic and epigenetic factors. Polymorphisms in genes that are implicated in the mechanisms of resistance listed above could be the main drivers of this inter-individual variability. For example, polymorphisms in *ERCC1* and *XRCC1*, implicated in nucleotide excision repair and base excision repair and therefore key in DNA repair mechanisms, have been associated with a better response in patients suffering NSCLC treated with Pt-compounds [27,28]. Another established example is the association between MDR1 polymorphisms in the Asian population suffering from NSCLC and resistance to PT-based compounds. The mechanism involved in this case would be a reduced intracellular drug accumulation [29]. Concerning some of the genes analyzed in our study, *MTHFR* rs1801133 genotype AA has been associated with an increased likelihood of response to chemotherapy when treated with Pt-compounds, as well as *UGT1A1* rs4148323 genotype AA which has been associated with a decreased response to cisplatin treatment in a cohort of East Asian patients with advanced NSCLC [22,30]. Nevertheless, these patients were treated with a combination of cisplatin and irinotecan, rendering the well-established correlation between *UGT1A1* polymorphisms and altered irinotecan pharmacokinetics the most probable cause of this response variation [31]. Our study did not find differences in response in patients with variations in *MTHFR* or *UGT1A1*.

This study is the first to propose that *DPYD* variant carriers are related to survival in a group of patients with NSCLC receiving Pt-based therapy, demonstrating a significantly poorer response in patients carrying these polymorphisms. Previous studies have shown that patients carrying the *DPYD* polymorphism have a lower response to Pt compounds in the treatment of other cancers. For example, Hongmei Zeng et al. described that *DPYD* variants are associated with an increased risk of poorer survival, no matter the treatment used (radiation therapy, surgery, 5FU, gemcitabine, erlotinib, capecitabine and cisplatin or oxaliplatin) in a cohort of patients with pancreatic cancer [32]. Another previous study carried out by Dhawan et al. reported that *DPYD* and/or *TPMT* variations were associated with reduced response rates and poorer OS, following a combination of cisplatin and 5-FU treatment in a cohort of 500 cases of patients of north Indian origin with head and neck squamous cell carcinoma [33]. *DPYD* encodes the enzyme dihydropyrimidine dehydrogenase (DPD), and mutations in *DPYD* can lead to reduced DPD activity, affecting the body’s ability to metabolize pyrimidine nucleosides and increasing the risk of toxicity following fluoropyrimidine administration. For this reason, genotyping for *DPYD* is a routine practice in daily clinical settings to aid decision-making regarding treatment with fluoropyrimidines [34,35,36]. Dhawan et al. discusses that carrying *TPMT* and/or *DPYD* polymorphisms may have increased levels of cisplatin and 5-FU that may have resulted in increased toxicity and therefore treatment discontinuation [33]. However, in our cohort of patients, only 1 of those with altered *DPYD* carried *TPMT* variations, and the patients did not experience greater Pt-related toxicity than patients with the wild-type genotype, suggesting other mechanisms may be involved in the treatment resistance observed in our cohort.

After entering the cell, Pt-compounds interact and form covalent bonds with different peptides and structures, including DNA. These interactions increase oxidative stress as the reducing equivalents in cells are depleted, causing harmful effects on the cells. However, these same molecules additionally serve as a protective buffer by neutralizing chemically active cisplatin and protecting DNA, becoming, therefore, a mechanism of Pt resistance [25]. In this regard, it has been suggested that *DPYD* deficiency is related to increased oxidative stress, which could serve as a possible explanation to the poor response observed in our patients in comparison to the rest of the sample [37]. Additional studies regarding genes related to *DPYD*, according to the STRING database (https://string-db.org/ accessed on 23 April 2023), such as *TYMP*, have additionally been linked to cisplatin resistance. Thymidine phosphorylase (TP), encoded by the *TYMP* gene, plays a role in the pyrimidine salvage pathway, and in vitro studies have shown that there might be a mechanism by which TP confers resistance to apoptosis induced by cisplatin [38].

Interestingly, although the DPD protein is not expressed in normal lung tissue (https://www.proteinatlas.org/ accessed on 23 April 2023), its RNA expression in a human lung cancer specimen has been associated with in vitro sensitivity to Pt-derived drugs [39]. Similar results have been identified in advanced rectal cancer patients treated with fluoropyrimidine-based chemotherapy, suggesting that *DPYD* expression confers a more aggressive behavior in these tumors [40,41]. Hence, it is evident that there exist disparities between *DPYD* in germline and tumor tissues, probably attributed to polymorphisms, epigenetic modifications or alterations in signaling pathways. Furthermore, other studies confirm that various epigenetic mechanisms, including DNA methylation in genes and ncRNAs that modify tumor cell survival, can be altered in patients with resistance to Pt chemotherapy [42,43,44]. Thus, additional studies are needed to further understand this matter, as it is probable that both genetic and epigenetic mechanisms are playing a joint key role in Pt treatment resistance.

In addition, our study suggests a possible association between the presence of variants in the *CYP2C8* gene and improved OS in patients treated with Pt-based therapy. We have not found other studies that support these findings. It is possible that the association between *CYP2C8* variations and cisplatin survival may not be direct, and other variables, such as the patient’s general health condition or the features of their tumor, may be responsible for this outcome. Additional investigation will be necessary to confirm and broaden these results.

Previous studies have described an association between ototoxicity caused by Pt compounds and variants in the *TPMT* gene [45,46]. However, results involving this association are inconsistent [47,48,49]. *TPMT* is involved in the metabolism of mercaptopurine and thioguanine; however, it additionally plays a role in the metabolism of cisplatin and other Pt-derived compounds [30]. In our study, we did not observe an association between *TPMT* variants and Pt-induced toxicity of any kind, possibly due to differences in study design, sample size or patient population. Other associations related to toxicity have been described, such as an increased risk of haematological adverse events in patients with variations determined in *UGT1A1* and *CYP2C8* [22,50]. Although these findings are not supported by our results, it is necessary to continue research to establish connections between genotype and toxicity to enhance the quality of life and improve treatment adherence in patients experiencing myelosuppression, nephrotoxicity and other limiting side effects such as periodontal disease [51].

Overall, our study shows the importance of considering genetic variability when designating and administering Pt-based chemotherapy and suggests that genetic testing may play an important role in future personalized cancer treatment. Understanding the mechanisms of therapy resistance is important for developing strategies to overcome treatment failure and improve patient outcomes. Although our findings show the potential application of *DPYD* genotyping as a predictive biomarker for cisplatin-based chemotherapy in NSCLC patients, future studies with a larger sample must be conducted to confirm these findings.

## 4. Materials and Methods

We conducted a retrospective exploratory study to identify pharmacogenetic biomarkers associated with the response to and toxicity of Pt-based drugs in patients with NSCLC. The inclusion criteria comprised patients with locally advanced or metastatic NSCLC, who were treated with Pt-based chemotherapy at La Paz University Hospital.

The pharmacogenetic test was performed at the local pharmacogenetic laboratory. Clinical data were collected from medical records. The maximum toxicity observed during the first 6 cycles of Pt treatment was recorded, according to the Common Terminology Criteria for Adverse Events version 5.0 (CTCAE v.5.0). Response was evaluated based on the first reevaluation CT scan and classified using the RECIST v1.1 criteria as follows: complete response, partial response, stable disease or progression.

Demographic characteristics including age and sex as well as a smoking habit, histologic type, tumor stage, chemotherapy regimen, Eastern Cooperative Oncology Group (ECOG) performance status scale, adverse effects, response in a reassessment CT at 3 months, radiotherapy and tumor progression were obtained from medical records and the laboratory information system. The median follow-up was 24 months. The study was approved by the hospital’s medical research ethics committee. Participants were informed in detail of the study’s aim and methods and afterwards provided written informed consent in accordance with the Declaration of Helsinki.

### 4.1. Genotyping Studies

Blood samples were obtained in a Vacutainer EDTA tube (Becton Dickinson, Franklin Lakes, NJ, USA). DNA was extracted from blood using a PureGene kit (Qiagen, Germantown, MD, USA), according to the manufacturer’s protocol. Pharmacogenetics tests were performed using OpenArray^®^ technology in the QuantStudio™ 12K Flex OpenArray^®^ System (Thermo Fisher Scientific, Waltham, MA, USA). Our pharmacogenetic test simultaneously analyses 60 single nucleotide polymorphisms (SNPs) in 14 genes, using TaqMan™ probes. Each TaqMan™ SNP assay contains two allele-specific probes and a primer pair to detect the specific SNP target (analyzed SNPs can be found in Appendix A). The validation study conducted for this technology was previously published [52].

The reaction was performed using the following settings: 10 min at 93 °C; 45 s at 95 °C, 13 s at 94 °C; and 2 min at 53.5 °C for 50 cycles, followed by 2 min at 25 °C. Data acquisition and analysis were carried out on the Thermo Fisher Cloud. Genotypes were inferred according to gene-specific guidelines, using a custom Java program.

### 4.2. Statistical Analysis

A descriptive analysis was conducted of our study population, including clinical and genetic variables. The variables were described using the number of participants (n), mean, standard deviation, minimum and maximum, median and interquartile range, depending on the type of distribution they followed. The normality of continuous variables was studied using the Kolmogorov–Smirnov test. In addition, association studies were conducted between the studied variables and their relationships with the collected information regarding the response to and toxicity of Pt-treatment.

Statistical comparisons were made between the analyzed genetic variants and the toxicity and efficacy outcomes. For this purpose, the Pearson’s chi-squared test was used to compare categorical variables. The main endpoint for evaluating the response to Pt-compounds based on different pharmacogenetic biomarkers was PFS, whereas the secondary endpoints were OS and the associations between pharmacogenetic variants and treatment-related toxicity.

The Kaplan–Meier method was used to estimate PFS and OS. For PFS, data from patients who were still alive and without disease progression or whose follow-up was lost were censored at the time of their last imaging evaluation. Events were considered as progression by a CT reassessment at 3 months. For OS, data from patients who were still alive or whose follow-up had been lost were censored at the last known time they were alive. The differences in OS and PFS between the two treatment groups were evaluated using the stratified logrank test. Hazard ratios and their corresponding 95% confidence intervals were estimated using a stratified Cox proportional-hazard model. Statistical analyses were carried out using SPSS 22.0 statistical software.

## 5. Conclusions

Our findings provide compelling evidence linking *DPYD* gene variants to platin resistance in NSCLC patients. These discoveries hold promising implications for personalized medicine, particularly in the optimization of chemotherapy. The identification of *DPYD* gene variants through such testing can serve as a valuable predictive biomarker, alerting healthcare professionals to potential resistance and empowering them to make well-informed decisions regarding treatment options. By establishing a significant association between *DPYD* gene variants and platin resistance in NSCLC patients, our study strongly advocates for the integration of genetic testing into clinical protocols, ultimately enhancing patient outcomes in the realm of platin-based chemotherapy.

## Figures and Tables

**Figure 1 ijms-24-09843-f001:**
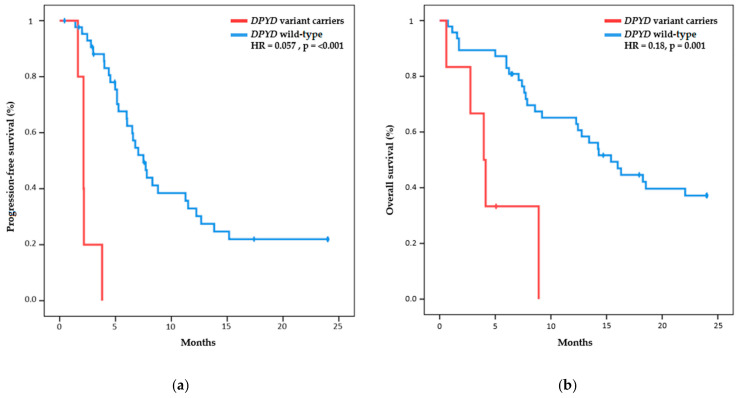
Survival curves: (**a**) Kaplan–Meier estimates of progression-free survival, according to germline *DPYD* variants; (**b**) Kaplan–Meier estimates of overall survival, according to germline *DPYD* variants.

**Figure 2 ijms-24-09843-f002:**
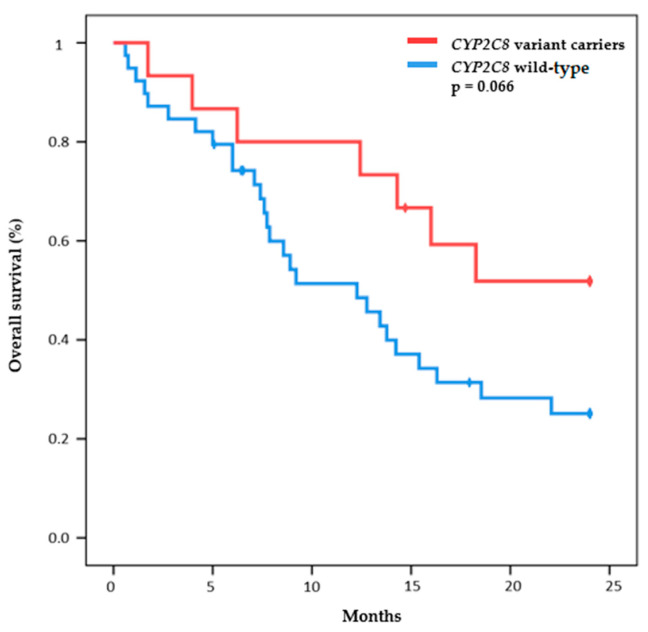
Overall survival curves according to CYP2C8 status.

**Table 1 ijms-24-09843-t001:** Demographic and Disease Characteristics of the Patients at Baseline.

	n = 55		
		Cisplatin-Based Chemotherapy (n = 22)	Carboplatin-Based Chemotherapy (n = 33)
Age (Average and range)	61 (25–81)		
Gender			
Female	20 (36%)	12	8
Male	35 (64%)	10	25
Histological type			
Adenocarcinoma	34 (60%)	15	19
Squamous cell carcinoma	17 (27.3%)	6	11
Other NSCLC	4 (12.7%)	1	3
Disease Stage			
IIIA	5 (9.1%)	4	1
IIIB	19 (34.5%)	9	10
IIIC	1 (1.8%)	0	1
IV	30 (54.5%)	9	21
Smoking status			
Non-smoker	4 (7.3%)	1	3
Ex-smoker	30 (54.5%)	12	18
Smoker	21 (38.2%)	9	12
ECOG performance status			
0	36 (65.5%)	12	24
1	16 (29.1%)	9	7
2	2 (3.6%)	1	1
3	1 (1.8%)	0	1

**Table 2 ijms-24-09843-t002:** Toxicity observed according to genotype analysis.

n: 55					
Gene	Status	Toxicity		Toxicity Grade ≥ 2	
		Yes	No	Yes	No
*UGT1A1*	WT *	21 (84%)	4 (16%)	6 (24%)	19 (76%)
	Variant carrier	23 (82.1)	5 (17.9%)	10 (35.7%)	18 (64.3%)
*MTHFR*	WT	21 (84%)	4 (16%)	7 (28%)	18 (72%)
	Variant carrier	23 (82.1%)	5 (17.9%)	9 (32.1%)	19 (67.9%)
*CYP2C19*	WT	21 (80.8%)	5 (19.2%)	17 (65.4%)	9 (34.6%)
	Variant carrier	23 (82.1%)	5 (17.9%)	7 (25%)	21 (75%)
*CYP3A5*	WT	37 (84.1%)	7 (15.9%)	15 (34.1%)	29 (65.9%)
	Variant carrier	7 (70%)	3 (30%)	1 (10%)	9 (90%)
*CYP3A4*	WT	39 (84.8%)	7 (15.2%)	13 (28.3%)	33 (71.7%)
	Variant carrier	5 (71.4%)	2 (28.6%)	3 (42.9%)	3 (57.1%)
*CYP2C9*	WT	25 (78.1%)	7 (21.9%)	9 (28.1%)	23 (71.9%)
	Variant carrier	19 (86.4%)	3 (13.6%)	7 (31.8%)	15 (68.2%)
*CYP2C8*	WT	32 (82.1%)	7 (17.9%)	14 (35.9%)	25 (64.1%)
	Variant carrier	13 (81.3%)	3 (18.7%)	3 (18.8%)	13 (81.3%)
*DPYD*	WT	39 (81.3%)	9 (18.8%)	15 (31.3%)	33 (68.8%)
	Variant carrier	1 (16.7%)	5 (83.3%)	1 (16.7%)	5 (83.3%)
*TPMT*	WT	41 (82%)	9 (18%)	17 (34%)	33 (66%)
	Variant carrier	4 (80%)	1 (20%)	0 (0%)	5 (100%)
*NUDT15*	WT	44 (81.5%)	10 (18.5%)	17 (31.5%)	37 (68.5%)
	Variant carrier	1 (100%)	0 (0%)	0 (0%)	1 (100%)
*TFAP2B*	WT	33 (86.8%)	5 (13.2%)	11 (28.9%)	27 (71.1%)
	Variant carrier	12 (70.6%)	5 (29.4%)	6 (35.3%)	11 (64.7%)
*EPAS1*	WT	25 (83.3%)	5 (16.7%)	8 (26.7%)	22 (73.3%)
	Variant carrier	19 (79.2%)	5 (20.8%)	8 (33.3%)	16 (66.7%)
*GM2A*	WT	42 (80.8%)	10 (19.2%)	15 (28.8%)	37 (71.2%)
	Variant carrier	3 (100%)	0 (0%)	2 (66.7%)	1 (33.3%)

* WT: wild-type. Results are expressed as frequencies with respect to the total number of patients with the wild-type genotype and patients carrying variants, respectively.

## Data Availability

The datasets generated during the current study are available from the corresponding author upon reasonable request.

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
