# Peer review of "Prognostic Impact of Dihydropyrimidine Dehydrogenase Germline Variants in Unresectable Non-Small Cell Lung Cancer Patients Treated with Platin-Based Chemotherapy"

_ijms, 2023, doi:10.3390/ijms24129843_

Round 1
Reviewer 1 Report
Dear authors,
The paper submitted by Guijarro-Eguinoa et al. approaches a very interesting topic which I found interesting and with a considerable practical impact.
I hope that my remarks and suggestions will be useful in order to increase the quality of the paper.
1. I noticed the high number of authors. Could you please clarify that?
2. For the corresponding authors make sure that you replace the personal e-mails with the professional ones;
3. Please carefully check the contributions of the two authors that have contributed equally as they have to be identical;
4. Line 196 – Discussion – Please kindly take into cosideration the impact of chemotherapy on the oral diseases like periodontal disease;
5. Line 340 – Conclusions section – I would suggest that you should take into consideration to rephrase this section in a more elaborated way that will reflect the novelty and clinical value of your findings in optimising future personalised clinical protocols.
Best of luck with your future research!
Dear Editor,
The paper submitted by Guijarro-Eguinoa et al. approaches a very interesting topic which I found interesting and with a considerable practical impact.
From my point of view the quality of English is good and doesn't require extensive revision.
I consider that the manuscript can be published after the recommended minor revisions.
Sincerely,
I. Luchian
Reviewer 2 Report
The manuscript discusses the prognostic potential of DYPD germline variants. The authors need focus stratifying a strong rationale for selecting DYPD. In addition authors need to perform validation experiments. Please find the comments below
1. The authors state, "In terms of molecular analysis, the genes with the most variants were UGT1A1 124 (52.83%), MTHFR (52.83%), CYP2C19 (51.85%), LY96 (47.17%), EPAS1 (44.44%), CYP2C9 125 (40.74%), TFAP2B (30.91%), CYP2C8 (29.09%) and CYP3A5 (18.52%). ". Why is DYPD not a part of this analysis?
2. The authors need to discuss what is the role of DYPD in lung cancer, in platinum based mechanisms. Also the authors can discuss if this gene has role in other cancers to add significance.
3. The authors need show what cancer mechanisms are impacted due to DYPD germline variants.
4. What approach was used to cal DYPD germline variant?
5. Table 2 does not display details on DYPD gene.
Overall, the manuscript looses focus and research interest. The authors need to validate their findings both bioinformatically and in vitro approaches.
The quality of English is good.
Round 2
Reviewer 2 Report
The authors have addressed all the concerns. I endorse the manuscript for publication.